# Attitude-Based Segmentation of Residential Self-Selection and Travel Behavior Changes Affected by COVID-19

**Chonnipa Puppateravanit \*, Kazushi Sano \* and Kiichiro Hatoyama**

Graduate School of Engineering, Nagaoka University of Technology, Nagaoka 940-2188, Japan; kii@vos.nagaokaut.ac.jp
\* Correspondence: s197012@stn.nagaokaut.ac.jp (C.P.); sano@nagaokaut.ac.jp (K.S.)

**Abstract:** This study evaluated the effects of COVID-19 on attitudes toward residential associated with travel behavior on decisions regarding future relocation. Chi-square automatic interaction detection was used to generate tree and classification segments to investigate the various segmentations of travelers and residents around mass transit stations. The decision tree revealed that the most influential variables were the number of transport card ownerships, walking distance to the nearest mass station, number of households, type of resident, property ownership, travel cost, and trip frequency. During the COVID-19 pandemic, people have concentrated on reducing travel time, reducing the number of transfers, and decreasing unnecessary trips. Consequently, people who live near mass transit stations less than 400 and 400–1000 m away prefer to live in residential and rural areas in the future. Structural Equation Modeling was used to confirm the relationship between attitudes in normal and pandemic situations. According to the findings, attitudes toward residential accessibility of travel modes were a significant determinant of attitudes toward residential location areas. This research demonstrates travelers' and residents' uncertain decision-making regarding relocation, allowing policymakers and transport authorities to better understand their behavior to improve transportation services.

**Keywords:** COVID-19 affected; residential self-selection; mass transit; decision tree; structural equation modeling; attitude-based

## 1. Introduction

The COVID-19 pandemic has effected several changes in a variety of fields such as economy, society, politics, government, population, and disease control management. COVID-19 has been detected all over the world for more than two years. People are becoming more aware of and concerned about pandemics; and their lifestyles, behavior, and attitudes are changing to avoid the spread of pandemics.

Most travel behavior studies have shown a significant decrease in travel, including avoiding the use of public transport and private cars. According to a study on changes in travel behavior due to the COVID-19 pandemic around the world, there was a significant shift from public transport (from 36% to 13%) to private transport (from 32% to 39%) and non-motorized modes (from 12% to 20%) during the pandemic [1]. The first wave of COVID-19 in Switzerland [2] found that it lowered the average daily distance traveled by a commuter by over 60% and the use of public transportation by more than 90%. Even in the short term, changes in travel behavior during the COVID-19 pandemic were evident.

Concentrating, on residential location analyses, transportation system resiliency, and longer-term aspects in pandemic situations should be considered in policy implementations and future planning [3]. Nevertheless, psychological factors have been demonstrated to be crucial in describing behavioral decisions more accurately for travel behavior studies. The attitudes might be related to the use of travel modes [4,5]. Consequently, travel attitudes and motives for relocation were examined and it was discovered that the reasons for

moving were related to travel [6]. However, housing and neighborhood characteristics are more important than travel-related attitudes, which influence travel behavior and residential choice [7].

In urban areas, mass transit is the most convenient and accessible mode of transport. Subway catchment areas, population and employment densities, land-use mix diversity, and intermodal connectivity have a positive impact on subway ridership [8]. The area around the mass transit station has been characterized differently from other areas by the surrounding infrastructure and the high accessibility it provides to commuters and residents near the stations. Urban travel characteristics indicate that the vast majority of inner-city residents (1) travel shorter distances than suburban residents [9], and (2) prefer traveling by train, indicating that people who moved closer to the stations became regular passengers [10].

Market segmentation is the identification of groups or market segments with similarities in characteristics or needs [11]. Market segmentation in travel behavior has been used to increase ridership, implement strategies/policies, improve services, etc. The segmentation of travelers can be based on multiple dimensions, such as identifying segments by different types of workers based on the predictability of their travel behavior over multiple days to understand changes in working patterns [12], or by commuting patterns to provide effective support for the planning and operation of public transport systems [13]. Recently, attitude-based market segmentation has found increased use in transportation research to get inside from a psychological perspective. According to a study on the attitude-based target group approach in predicting the ecological impact of mobility behavior, the results showed that the predictive power of the attitude-based approach was higher than that of segmentation based on sociodemographic and geographic factors [14].

This study examines the relationship between residential location and travel mode behavior as impacted by attitudes toward relocation, as well as the impact of COVID-19, to understand the tendency of behavior in the future. The objectives of this study are as follows:

1. To investigate the impact of COVID-19 on behavior and attitude by studying attitudes toward relocation of (1) residential location area, and (2) residential accessibility, on the travel mode associated with travel behavior, which leads to future relocation decisions.
2. To identify and categorize the segmentation of characteristics of travelers and residents around the mass transit station area, based on attitude change in (1), the short-term decision of attitude toward (1a) residential accessibility of the travel mode, and (1b) concern for using public transportation, and (2) the long-term decision of attitude toward (2a) residential location area, and (2b) concern for living in an urban area.
3. To confirm the relationship between the effects of attitudes toward residential accessibility, and the residential location areas, pre-test and post-test designs were applied to investigate the relationship of intervention variables from the COVID-19 pandemic.

In this study, the Chi-square Automatic Interaction Detection (CHAID) algorithm was applied to classify travelers and residents into segment groups based on the multi-way splits algorithm for building a decision tree; and separate characteristics of travelers and residents into groups under attitude toward relocation and provide a more in-depth understanding of the effect the COVID-19 phenomenon had on the case study. Structural Equation Modeling (SEM) was used to confirm the relationship between attitudes that were affected before and during the COVID-19 period, and the consequent model (pre-test and post-test) illustrated the phenomenal effect of COVID-19.

## 2. Literature Review

### 2.1. Residential Self-Selection and Attitude

There has been a debate regarding residential self-selection or relocation in past transportation research, which was marked by an objective-subjective division in understanding travel behavior [4,15]. Hard factors such as urban form and socioeconomic factors are

recognized as having an impact on various aspects of travel behavior. Travel behavior research also used soft factors [4], such as attitudes and preferences for various modes of transportation or neighborhood characteristics [16], and evaluates their impact on travel behavior. Additionally, personal characteristics and travel-related attitudes were found to be significant predictors of how people prefer to travel [17].

Considering the factors of residential location related to travel behavior, the availability of public transit is the most important factor influencing current residential location choices, followed by living in a good neighborhood and housing affordability [18]. Nevertheless, the type of residential location had little effect on travel behavior, whereas attitude and lifestyle variables had an outstanding impact on travel demand [19]. Furthermore, the relationship between changes in the built environment, changes in auto ownership, and changes in travel behavior revealed that relocating to neighborhoods closer to destinations or with alternative transport options could result in less driving and more walking [20]. This is evident in residential self-selection, which includes neighborhood preferences and/or travel-related attitudes, as well as the built environment and socio-demographic characteristics, all of which have a significant impact on travel behavior.

Additionally, relocations and associated changes in the built environment cause significant changes in car ownership and travel mode usage, as well as changes in household structure that tend to accompany relocations, which have significant effects [21]. Moreover, the mode of travel was shown to be associated with residential relocation, with statistically significant relationships between modal shifts and selected explanatory factors. However, the important factors when deciding to change from private car to public transit were car ownership, additional car purchase, income, specific housing type and size, relocation type, the convenience of the subway/bus for commuting, change in commuting distance, and distance to subway station [22].

### 2.2. Decision Tree on Travel Behavior Research

A decision tree is an intuitive, easy-to-implement, and productive modeling technique that can be depicted as a tree for classifying customers [23]. Recently, decision trees have been used in decision-making processes, and have been demonstrated to be an effective approach for making decisions. The decision tree for classification has four algorithms: Classification and Regression Trees (CART), exhaustive CHAID, CHAID, and Quick, Unbiased, Efficient, and Statistical tree (QUEST) [24]. This study addresses CART and CHAID, which represent classification and regression trees and use nonparametric statistical techniques that can be used for categorical and continuous data.

The CART, first presented by Gordon et al. (1984), uses a binary tree technique based on the sum of squared estimates of errors between the observation and the mean value of the node, and the Gini diversity index as a measure of impurity when deciding to split. However, the CART always produces binary trees, and the binary tree is not an efficient representation and can be difficult to interpret [25]. CHAID, proposed by Kass (1975) [26], is a decision tree technique based on the chi-squared test when determining the best splitting pattern for tree classifiers. CHAID has been used for the prediction, classification, and detection and establishment of relationships between variables. CHAID decision trees use nonparametric techniques that make no assumptions about data and are mostly used in market research for segmentation.

In transportation research, some studies used CART and CHAID in association with logistic regression to classify attribute variables more precisely, such as applying CART analysis to obtain the attribute levels of comfort, speed, and travel cost, which proved to be efficient for later applications [27]. According to the study, Jang and Ko (2019) used CHAID analysis to obtain commute time ranges with significantly different compositions of satisfied and unsatisfied commuters by dividing the dataset based on the travel time range [28]. Levin and Zahavi (2001) studied CHAID using the logistic regression model as a benchmark and found that automatic segmentation methods may substitute judgmentally based segmentation methods for response analysis [29]. In a study of travel behavior mod-

els, CHAID was also investigated using segmentation analysis and was used to examine the rates of household trip generation. The model's predictive capability was verified, and the results suggested that CHAID can be used as an exploratory technique to aid model development, or as a model in and of itself [30]. In addition to the trip distribution model, CHAID applied traditional gravity models to estimate destination choices and compared them to the decision tree (CHAID and CART) approaches. The results show that the CHAID algorithm produced the best fit for real destination choices. They suggested that decision tree algorithms could be used in distribution modeling to improve traditional trip distribution approaches by assimilating the effects of disaggregated variables [31].

Therefore, to determine the most effective and efficient ways to investigate how different segments affect attitudes toward relocation, COVID-19 concerns of travelers and resident decision-making, the chi-square automatic interaction detection (CHAID) method is one of the most effective segmentation approaches. In this study, the CHAID algorithm was applied because it allows multi-way splits for nodes and is more flexible when used with category variables that are suited for the study of segmentation of characteristics under consideration of attitude-dependent variables.

### 2.3. Structural Equation Modeling on Residential Self-Selection

SEM is a statistical technique for testing and estimating causal relationships. The purpose of SEM is to test and develop theories. SEM is generally considered a confirmatory procedure rather than an exploratory one [32]. The analysis of paths and factors is the basic concept and origin of structural equation modeling. Thus, a summary of the structural equation model is presented. This is the outcome of the synthesis of three major data analysis techniques: factor analysis, path analysis, and regression analysis [33].

In transportation research and residential self-selection, SEM was used in the correlation analysis of the impact of travel behavior and residential relocation. For example, SEM was used to investigate the relationship between land use and travel patterns that impact weekend travel compared with workday travel. They showed that land use has an opposing effect on travel mode choice and trip frequency on weekdays compared to weekend travel [34]. Nonetheless, changes in neighborhood characteristics lead to changes in travel choices, and neighborhood characteristics influence travel behavior and have an additional impact on travel behavior through their influence on automobile ownership [35]. Moreover, the relationship between the built environment and travel attitude in travel behavior was used in the SEM to estimate the residential self-selection and environment determination frameworks. The results argue that both residential self-selection and residential determination are defined by the complex relationships between the built environment, travel attitude, and travel behavior [36].

### 2.4. COVID-19 on Travel Behavior Change

The COVID-19 epidemic has begun to have a significant impact on people's lives worldwide, affecting their behavior in both the short and long term, having both physical and mental impacts. People will reduce their travel due to COVID-19 and will prefer to use active modes or cars over public transit [1]. In the short term, owing to pandemic control and various measures, as well as the curbs on public transport services, workday travel behavior will gradually change the commuters' decision-making regarding their travel behavior because of COVID-19 prevention measures such as physical distancing. In India, 41.65% of people stopped traveling during the transition to lockdown period, while 51.31% continued to use the same mode of transportation as previously [37].

The pandemic has had a major impact on public transport due to concerns about being in contact with, or close to people at risk of infection, and policy responses to disease control. Regarding the level of hygiene on public transportation, it was found that 58% of passengers have been more concerned about it post-COVID-19 than earlier [38]. Evidently, people are concerned about using the public transport system and their travel intentions have been disturbed. The first wave of COVID-19 in Switzerland reduced the average

commuting distance by approximately 60% and public transport usage by over 90% [39]. Additionally, the huge average decreases in travel and public transport usage as a result of the pandemic and associated policy responses mask major differences across socioeconomic groups, with the average travel decreasing less among the less educated and lower-income groups [40]. According to a study on public transport use in the United States, lower-income transit passengers reduced their travel less than others who were unwilling to use transit because of the risk of infection. However, mask usage and reducing crowding may increase transport users' willingness to utilize it [41]. People's preferences for housing types may change as a consequence of COVID-19, and the quality of living environments will almost certainly become a significant factor [42].

## 3. Descriptive Statistics

### 3.1. Data Collection

This study focused on mass transit station areas. Bangkok, Thailand, was selected for this study. Note that after the mass transport system was implemented in Bangkok, 77 percent of citizens changed from private cars to mass transit [43]. The study area was located around the existing mass transit station area to focus on the target group of travelers and residents around the station, which represents an area of easy access to mass transit. A map of the survey area with the existing mass transit stations is shown in Figure 1. The survey catchment area consisted of all within 1000 m from a mass transit station. According to a previous study conducted in Bangkok [44], the proportion of people walking to the stations decreased when the distance to the station was more than 400 m, while less than 10% of people walked more than 1 km to the station because long distance is associated with a lower probability of walking to public transportation [45]. According to the study area, existing mass transit stations are mainly located in the Bangkok area, with some stations in Nonthaburi and Samut Prakan.

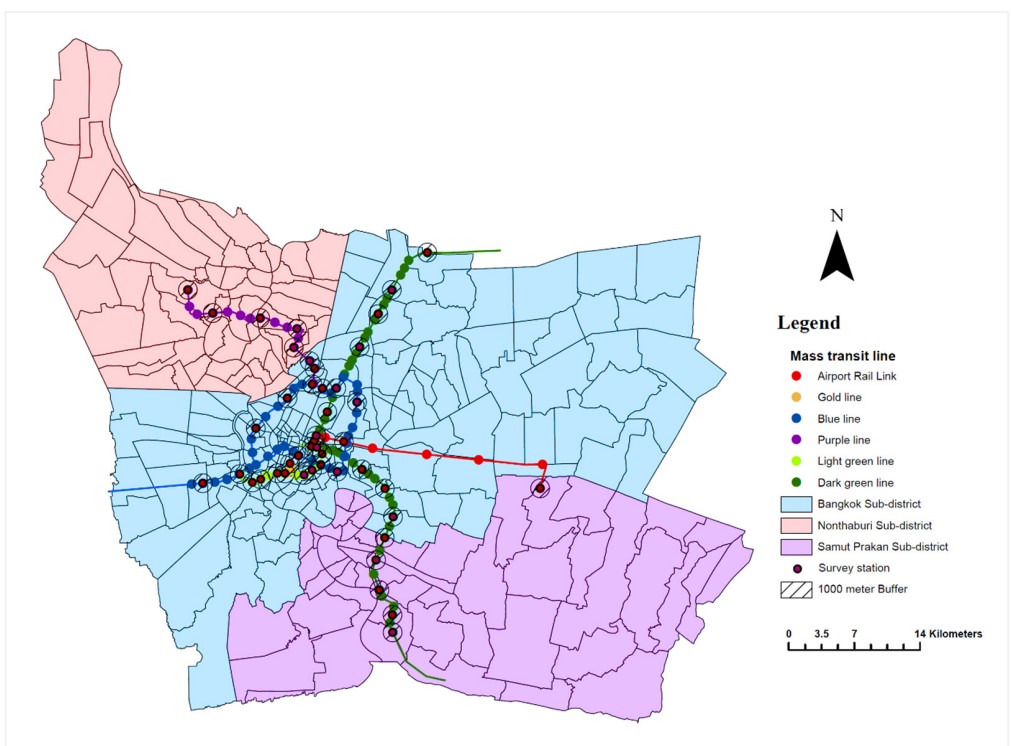

**Figure 1.** Study area of existing mass transit station and survey area.

The survey was conducted in the Bangkok metropolitan area in December 2020, covering all existing mass transit stations in the area. At that time, there were six lines of mass transit in operation, including the BTS light green (54.3 km), BTS dark green (14 km),

MRT blue line (47 km), MRT purple line (23 km), airport rail link (28.5 km) [46], and gold line (1.74 km) [47], for a total of 168.54 km and 125 stations. Nevertheless, during the COVID-19 pandemic, Thailand's Department of Rail Transport disclosed that the ridership of mass transit decreased by approximately 80% in April (the first wave of COVID-19) compared with January 2020 [48].

At the time of the survey, during the COVID-19 situation, there were no lockdown restrictions, but a state of emergency was maintained. However, the questionnaire survey was conducted face-to-face with social distancing. Considering the context of the country, the online questionnaire had a low response rate and could lead to selection bias for young people, those who can access the Internet, and people who are familiar with the online survey. The questionnaire had four major sections: (1) personal characteristics, (2) changes in sociodemographic and travel behavior, (3) trip characteristics, and (4) attitudes toward relocation (attitude toward residential location area: opinion regarding three statements; attitude toward residential accessibility: opinion regarding three statements) and attitude toward COVID-19 concern: two statements. Respondents were asked to compare their attitudes to the situations before and during COVID-19 in Sections 2–4. In addition, attitude factors were collected by using a 5-point Likert scale (5 = strongly agree to 1 = strongly disagree).

*3.2. Sample Characteristics*

This study focused on commuting trips on weekdays. In all, 682 complete responses were collected for analysis in this study. Table 1 contains statistical information about the respondents, including sociodemographic, residential, and traveler characteristics.

According to the sociodemographic characteristics of the responses, the respondents were split as 63% women and 37%. Men. The majority of the respondents were between the age groups of 18–24 and 25–26 years old (25% and 26%, respectively), with 42% having bachelor's degrees and 32% having high school education. Most of them were employed (58%), including government officials, state enterprise employees, and private company employees.

In terms of Bangkok's population in 2020, the total population was 8,854,718, of which, women made up 52 percent [49]. However, the population represented in this research consists of residents and travelers in the area of the mass transit station. This may be a different circumstance in relation to Bangkok's general population. In the previous research on the demographics of people residing in the station area, respondents with comparable characteristics were also uncovered. It was discovered that the majority of the respondents were female (62.8 percent), with 58.7 percent of car ownership [50].

Regarding the residential characteristics of the respondents, it was found that the majority lived with 2–3 people (30% and 26%, respectively), in apartments (33%), and most preferred to live in single houses (38%). People who lived in rented houses were 47% and those who owned houses were 45%. Consequently, 45% are not paying for housing costs per month. In terms of travel characteristics, 50% of respondents had no vehicle, 32% had one vehicle, 60% had no transport card, and 31% had one transport card.

Transport cards have been operated independently by private operators for mass transit systems because the mass transit system authority was unsuccessful in integrating transport card systems in 2020. It is possible for the respondent to carry more than one card if a transfer from one transit system to another is required. The card types included MRT, MRT Plus, Rabbit, Smart Pass, and Mangmoom cards.

**Table 1.** Characteristics of respondents.

| Description | Variable | Category | Percentage (n) |
|---|---|---|---|
| Sociodemographic characteristics | | | |
| Gender | S01 | Male | 37% (249) |
| | | Female | 63% (433) |
| Age | S02 | <18 years | 2% (17) |
| | | 18–24 years | 25% (172) |
| | | 25–34 years | 26% (176) |
| | | 35–44 years | 18% (120) |
| | | 45–54 years | 15% (99) |
| | | 55–64 years | 10% (71) |
| | | >64 years | 4% (27) |
| Education | S03 | <High school | 6% (39) |
| | | High school | 32% (220) |
| | | College | 17% (117) |
| | | Bachelor's degree | 42% (288) |
| | | ≥Master's degree | 3% (18) |
| Occupation | S04 | Student | 17% (120) |
| | | Employee | 58% (393) |
| | | Personal Business | 14% (93) |
| | | Unemployed | 10% (66) |
| | | Other jobs | 1% (10) |
| Residential characteristics | | | |
| | R01 | 1 person | 12% (81) |
| | | 2 persons | 30% (205) |
| No. of people in a household | | 3 persons | 26% (176) |
| | | 4 persons | 17% (115) |
| | | ≥5 persons | 15% (105) |
| Type of residence | R02 | Apartment | 33% (228) |
| | | Condominium | 8% (55) |
| | | Single house | 38% (261) |
| | | Townhouse | 14% (94) |
| | | Other | 7% (44) |
| Property ownership | R03 | Hire purchase | 8% (50) |
| | | Owner | 45% (310) |
| | | Rent | 47% (322) |
| Housing cost/month | R04 | <3500 THB | 10% (67) |
| | | 3501–5000 THB | 27% (183) |
| | | 5001–7500 THB | 11% (75) |
| | | 7501–10,000 THB | 4% (25) |
| | | 10,001–15,000 THB | 2% (17) |
| | | 15,001–20,000 THB | 1% (6) |
| | | 20,001–30,000 THB | 0% (2) |
| | | 30,001–50,000 THB | 0% (0) |
| | | >50,000 THB | 0% (1) |
| | | No pay | 45% (306) |
| Traveler characteristics | | | |
| | T01 | No vehicle | 50% (339) |
| | | 1 car | 32% (220) |
| No. of vehicle ownership | | 2 cars | 13% (93) |
| | | 3 cars | 3% (18) |
| | | ≥4 cars | 2% (12) |
| | T02 | No card | 60% (414) |
| | | 1 card | 31% (211) |
| No. of Transport card ownership | | 2 cards | 8% (53) |
| | | ≥3 cards | 1% (4) |
| | T03 | <400 m | 29% (202) |
| Walking distance to nearest station | | 400–1000 m | 44% (298) |
| | | >1000 m | 27% (182) |

### 3.3. Behavior Change

The survey revealed that the COVID-19 pandemic has resulted in changes in sociodemographic and travel behaviors. As seen in Table 2, the change in sociodemographic income before and during COVID-19 showed an increase in the number of people in the income range of 0–18,000 THB (0–600 USD) per month by 3%, which shows the overall income affected by COVID-19. The number of people in the middle and high-income range of >18,000 THB (>600 USD) per month decreased from pre-COVID-19 in total by 3%. Note that the average household income per month in the Bangkok Metropolitan Region in 2019 (pre-COVID-19) was 37,751 THB (1256.48 USD) [51]. When comparing with the pre-COVID-19 pandemic period, it was discovered that commuting to work at an office or factory reduced by 2%, and overall work outside the home reduced by 4%, which corresponds to an increase in work from home by 4%. No differences were observed for other places/workplaces compared with the pre-COVID-19 period.

**Table 2.** Characteristics of behavior changes of respondents.

| Description | Category | Pre-COVID-19 | | During COVID-19 | |
|---|---|---|---|---|---|
| | | Variable | Percentage (n) | Variable | Percentage (n) |
| Change in sociodemographic | | | | | |
| | <7500 THB | S15 | 15% (102) | S25 | 16% (110) |
| | 7501–18,000 THB | | 42% (286) | | 44% (298) |
| | 18,001–24,000 THB | | 22% (150) | | 21% (142) |
| Income/month | 24,001–35,000 THB | | 13% (88) | | 12% (82) |
| | 35,001–50,000 THB | | 4% (28) | | 4% (26) |
| | 50,001–85,000 THB | | 2% (18) | | 2% (17) |
| | 85,001–160,000 THB | | 1% (6) | | 0% (3) |
| | >160,000 THB | | 1% (4) | | 1% (4) |
| | Office/Factory | S16 | 56% (387) | S26 | 54% (363) |
| | Home | | 9% (61) | | 13% (87) |
| Place of work | Coffee shop | | 2% (12) | | 1% (10) |
| | Field site | | 2% (11) | | 1% (10) |
| | Co-working space | | 0% (1) | | 0% (1) |
| | Other/no | | 31% (210) | | 31% (211) |

### 3.4. Travel Behavior Change

The survey was divided into two parts: travel characteristics before the pandemic (pre-COVID-19) and travel characteristics during the pandemic (during-COVID-19). Changes in travel behavior were obtained from the responses to trip characteristics in the survey to explain daily trips (one-way trips) on weekdays or usual trips. It was shown that most people travel 4–6 trips per week (65%) and only 20% traveled 0–3 trips per week. However, people reduced overall weekly trip frequency during the COVID-19 pandemic, resulting in a 5% increase in the 0–3 trips per week category, compared with before the outbreak. From the responses to the number of trips per day, it was found that 93% traveled 0–2 trips per day during the pre-COVID-19 and 94.6% during the COVID-19 pandemic, with the number of trips per day decreasing from 7% to 6% (see Table 3).

Respondents' commute trips generally necessitate transfers within the transport mode or between multiple modes to get to the destination. According to the results of the survey, 44% of the trips in one day required transfers 4–5 times per day, while 37% required 6–7 times per day, in the pre-COVID-19 period. During the pandemic, people tried to reduce travel and mode of transfer, and the number of transfers in the 2–3 transfer times per day category increased by 2%. Respondents who spent 31–60 min on all commuting trips per day were 31%, while those who spent 61–90 min on all commuting trips per day were 26% in the pre-COVID-19 period. During the COVID-19 period, people who spent time traveling more than 60 min on all commuting trips per day reduced their time by 3%. Hence, people who travel less than 60 min on all commuting trips per day

increased by 3%. As a consequence of overall travel time, respondents who were spending 51–100 THB (1.67–3.33 USD) per day were 50% and those spending 0–50 THB (0–1.67 USD) per day were 28%. However, during the COVID-19 period, people who spent more than 50 THB (>1.67 USD) per day on travel reduced by 3%, whereas those who spent 0–50 THB (0–1.67 USD) per day on travel increased by 3%.

**Table 3.** Characteristics of behavior changes respondents. (Cont.).

| Description | Category | Pre-COVID-19 | | During COVID-19 | |
|---|---|---|---|---|---|
| | | **Variable** | **Percentage (n)** | **Variable** | **Percentage (n)** |
| Change in travel behavior | | | | | |
| Trip frequency | 0–3 trips/week | T14 | 20% (139) | T24 | 25% (171) |
| | 4–6 trips/week | | 65% (442) | | 61% (418) |
| | 7–9 trips/week | | 4% (29) | | 4% (24) |
| | ≥10 trips/week | | 11% (72) | | 10% (69) |
| Number of trips | 0–2 trips/day | T15 | 93% (634) | T25 | 94% (642) |
| | 3–4 trips/day | | 7% (46) | | 6% (38) |
| | ≥5 trips/day | | 0% (2) | | 0% (2) |
| Number of transfers | 0–1 times/day | T16 | 0% (0) | T26 | 0% (0) |
| | 2–3 times/day | | 13% (85) | | 15% (100) |
| | 4–5 times/day | | 44% (302) | | 44% (301) |
| | 6–7 times/day | | 37% (251) | | 36% (246) |
| | 8–9 times/day | | 6% (41) | | 5% (35) |
| | ≥10 times/day | | 0% (3) | | 0% (0) |
| Travel time | 0–30 min/day | T17 | 7% (50) | T27 | 8% (57) |
| | 31–60 min/day | | 31% (212) | | 33% (227) |
| | 61–90 min/day | | 26% (179) | | 25% (167) |
| | 91–120 min/day | | 17% (117) | | 16% (111) |
| | 121–180 min/day | | 13% (87) | | 13% (86) |
| | >180 min/day | | 6% (37) | | 5% (34) |
| Travel cost | 0–50 THB/day | T18 | 28% (193) | T28 | 31% (209) |
| | 51–100 THB/day | | 50% (338) | | 48% (327) |
| | 101–150 THB/day | | 15% (99) | | 14% (96) |
| | >150 THB/day | | 7% (52) | | 7% (50) |

For commuting or usual trips, there may be more than one purpose for the trips. The main purpose of travel, in this study, is divided into six categories (1) school/work (SW), (2) shopping/eating/exercise (SH), (3) visit (VS), (4) personal business (PB), (5) home (HM), (6) other (OT). According to the survey results, 94% of the sample was traveling for one trip purpose, with 74% of respondents commuting mainly for work or school, with an approximately 2% reduction in travel during the COVID-19 pandemic. Shopping and recreation trips in the pre-COVID-19 period were 10%, which increased to 12% during the pandemic. Travel for two purposes per day (6% of the sample) changed slightly compared to the pre and during COVID-19 periods, such as commuting to work/school with shopping/eating/exercise purposes (SW + SH + HM), work/school with a personal business purpose (SW + PB + HM), and other with shopping/eating/exercise purposes (OT + SH + HM). However, there is no difference in traveling for the three purposes per day (see Figure 2).

Referring to the survey area, available travel modes were divided into 18 modes from the questionnaire that covered all transport modes in the Bangkok metropolitan area. Considering the transportation accessibility characteristics, the traveling modes are divided into five categories as follows: (1) non-motorized (NM), including walking and bicycle; (2) Motorized (MO), including private car and motorcycle; (3) Paratransit (PR), including motorcycle taxi and private car taxi; (4) Feeder transit (FD), including bus, BRT, passenger van, Chao Phraya Express boat, Khlong boat, and local train; (5) Mass transit (MT), including BTS dark green line, BTS light green line, MRT blue line, MRT purple line, ARL airport rail link, and monorail gold line.

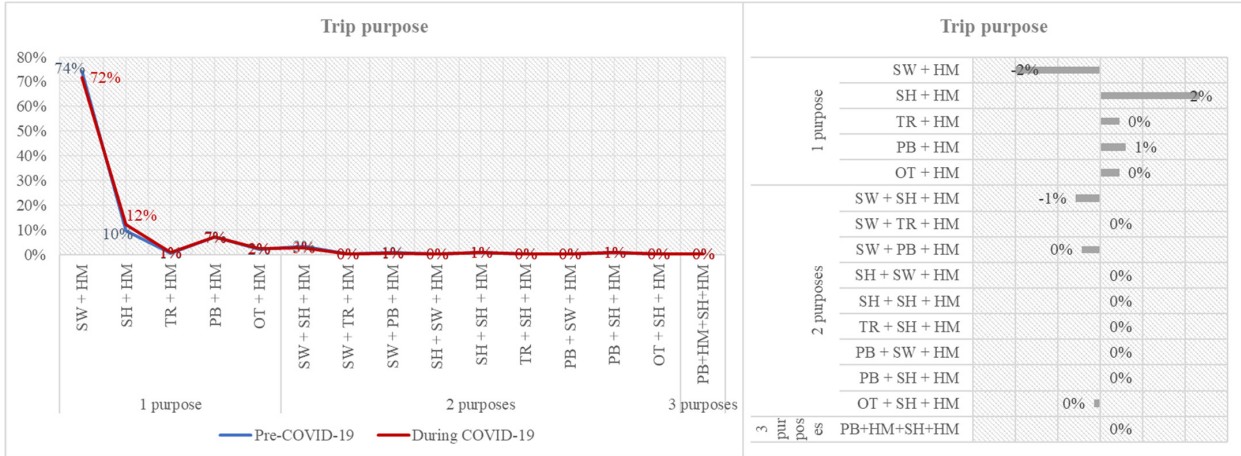

**Figure 2.** Comparison of trip purpose in pre-COVID-19 (TP1) and during COVID-19 period (TP2).

In the pre-COVID periods, 45% of the respondents used only one mode of travel, 20% used mass transit, and 14% used feeder transit. However, during the COVID-19 period the number of respondents using non-motorized modes increased from 2% to 3%, and those using paratransit by 1%. A total of 47% of the respondents use two modes of travel per day pre-COVID-19, which decreased to 45% during COVID-19. This demonstrated that people attempted to minimize their travel and transfer modes as much as possible to minimize meeting people while traveling and reduce their chances of contracting COVID-19. Respondents who preferred to travel by personal vehicles (motorized) and mass transit were 24% and 23%, respectively, during the COVID-19 period. Feeder transit and mass transit were used by 15% of respondents pre-COVID-19, and by 14% during COVID-19. The three modes of trip preference did not change before and during the COVID-19 outbreak, 8% of respondents traveled using three modes per day. This would be because these travelers do not have many options for their commute and found that among these, those who used motorized, feeder transit and mass transit were 7%, while those who traveled by motorized, paratransit, and mass transit were 1% of respondents. The details of mode share are shown in Figure 3.

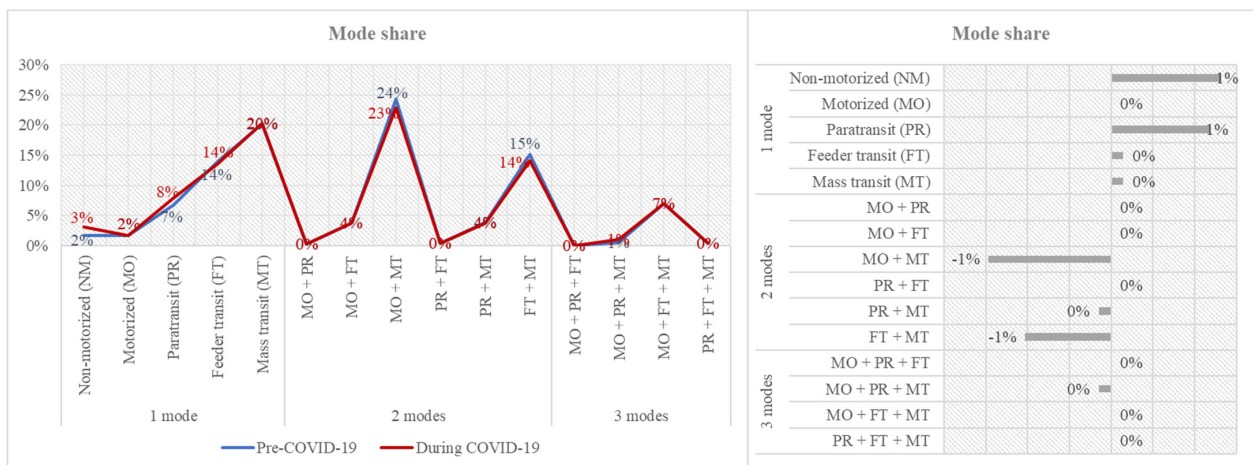

**Figure 3.** Comparison of mode share in pre-COVID-19 (TM1) and during COVID-19 period (TM2).

*3.5. Attitude Change*

This study considered attitude questions to quantify the effects of attitudes related to relocation and travel behavior on residents and travelers near mass transit stations. The attitude change factor affected by COVID-19 was collected, including responses to

eight statements: attitude toward residential location area (three statements), attitude toward residential accessibility (three statements), and attitude toward the concern of COVID-19 (two statements), divided into pre and during COVID-19 questions. All attitudes were considered using a 5-point Likert scale (5 = strongly agree to 1 = strongly disagree). Reliability analysis for attitude was between 0.78 and 0.96, as shown in Table 4. However, a Cronbach's alpha value of more than 0.7 is generally accepted on a moderately to excellently reliable scale.

**Table 4.** Attitude in pre-COVID-19 and during COVID-19 period.

| Factor | Pre-COVID-19 | | | | During COVID-19 | | | |
|---|---|---|---|---|---|---|---|---|
| | Variables | $\alpha$ [1] | Median | SD | Variables | $\alpha$ [1] | Median | SD |
| Attitude toward residential location area | | | | | | | | |
| Prefer to live in urban area. | 1UrbanArea | 0.798 | 3 | 0.78 | 2UrbanArea | 0.790 | 4 | 0.80 |
| Prefer to live in residential areas. | 1ResidentialArea | 0.797 | 4 | 0.80 | 2ResidentialArea | 0.791 | 4 | 0.86 |
| Prefer to live in rural area. | 1RuralArea | 0.810 | 4 | 0.91 | 2RuralArea | 0.806 | 4 | 0.94 |
| Attitude toward residential accessibility | | | | | | | | |
| Prefer residential area near mass transit station. | 1MassTransit | 0.793 | 4 | 0.80 | 2MassTransit | 0.789 | 4 | 0.87 |
| Prefer residential area near bus stop. | 1BusStop | 0.792 | 3 | 0.84 | 2BusStop | 0.789 | 4 | 0.90 |
| Prefer residential area near highways or main roads. | 1Highway | 0.795 | 4 | 0.95 | 2Highway | 0.791 | 4 | 0.93 |
| Attitude toward concern of COVID-19 | | | | | | | | |
| Not choosing to live in an urban area due to concern about infection. | 1UrbanConcern | 0.802 | 3 | 0.85 | 2UrbanConcern | 0.797 | 4 | 0.91 |
| Worried about infection concerns to use public transport. | 1PTconcern | 0.808 | 3 | 0.88 | 2PTconcern | 0.802 | 4 | 0.96 |

[1] Cronbach's Alpha.

The result of the attitude toward residential location areas found that people prefer to live in urban areas during the COVID-19 period, while residential areas and rural areas have no difference in attitude between pre and during the COVID-19 period. Considering that during a pandemic, it is difficult to access hospitals due to hospital congestion and limited medical personnel, there is a possibility that people prefer to live in an urban area with easier access to hospitals and grocery stores. Conversely, in response to attitude toward residential accessibility, most respondents preferred residential areas near bus stops. However, there was no change in the preferred residential areas near mass transit stations and highways or main roads before and after COVID-19. Nevertheless, the attitude toward concern about COVID-19 was found to be more significant in terms of the respondents' choice not to live in an urban area or preferring not to use public transport due to concerns about infection.

## 4. Decision Tree Analysis

In this study, the decision on attitude change was analyzed using decision trees of the CHAID algorithm to identify the segmentation of travelers and resident characteristics near a mass transit station. The CHAID algorithm was analyzed using IBM SPSS version 26 to develop a decision tree. CHAID's algorithm performs a sequence of merging, splitting, and stopping processes based on user-defined criteria such as the chi-square test significance level, minimum node or segment size, and maximum tree depth level [28]. For the CHAID in this study, the specification for developing a tree uses user-specified split model criteria, including: (1) the significance level for splitting nodes and merging is set at 0.05; (2) the number of cases for parent nodes is limited to 50; and (3) the minimum number of instances for a child node is set at ten. The maximum tree depth was controlled by the minimum segment size. A 10-fold cross-validation approach was used to estimate the misclassification risk of the model. The accuracy and detection of CHAID are represented as a percentage.

The research hypothesis was to explore the characteristics of travelers and residents around mass transit stations during COVID-19 conditions, and the relationship between the independent variables is at each level of the dependent variable. The dependent variables were determined using the attitude factor. Eight factors were applied to each model. However, to consider attitudes in a positive and negative way due to consideration of segment analysis, a dichotomous choice was applied. This scale allows for the determination of the level of agreement or disagreement among the respondents. A 5-point Likert scale (5 = strongly agree to 1 = strongly disagree) was transferred to positive (scale 5 and 4) and negative scales (scale 3, 2, and 1). The model divided the pre-COVID and during COVID-19 using a single decision tree with a total of 20 independent variables and one dependent variable.

### 4.1. Segmentation by Attitude toward Residential Location Area

The CHAID tree of attitudes toward residential location areas was divided into three models: (1) prefer to live in urban areas, (2) prefer to live in residential areas, and (3) prefer to live in rural areas. The tree analysis results in Table 5 show the relevant segmentation of attitudes toward residential location areas and the decision rules for terminal nodes. The results of the attitude toward residential location areas demonstrate that the number of transport card ownerships was the most important variable in splitting segments in attitude toward preferring to live in urban areas pre-COVID and during the COVID-19 period. However, attitudes toward preferring to live in urban areas in the pre-COVID-19 period found that travel costs and property ownership were given more priority than during COVID-19. Conversely, during the COVID-19 pandemic, it was found that walking distance to the nearest station, place of work, and trip frequency were more important than pre-COVID-19. For attitudes toward preferring to live in residential areas, travel cost/day was found, and the number of transport card ownerships was the most important variable for the splitting segment on pre-COVID-19 and during COVID-19, respectively. The variable difference between attitudes toward preferring to live in urban and residential areas is education and the type of residence, which is related to those who prefer to live in residential areas. Additionally, the attitude toward preferring to live in rural areas in the pre-COVID-19 period revealed trip frequency to be a more important variable. Meanwhile, during the COVID-19 period, it was more important to consider the type of residential.

### 4.2. Segmentation by Attitude toward Residential Accessibility

The CHAID tree of attitude toward residential accessibility was divided into three models: (1) prefer residential areas near mass transit stations, (2) prefer residential areas near bus stops, and (3) prefer residential areas near highways or main roads. The tree analysis results in Table 6 present the relevant segmentation of attitudes toward residential accessibility and the decision rules for terminal nodes. Overall, trip frequency and the number of transport cards owned were the most important variables in the split segment in attitude toward residential areas near mass transit during the pre-COVID and COVID-19 periods, respectively. Nonetheless, during COVID-19, it was shown that walking distance to the nearest station, type of residence, number of cars owned, and travel time were all significant variables. Interestingly, regarding the attitude toward preferring residential areas near bus stops pre-COVID-19, the number of households was not important. Furthermore, during COVID-19, the number of households, education, gender, and the number of transfers were significantly different from pre-COVID-19. Although attitudes toward preferred residential areas near highways or main roads were explored, the walking distance to the nearest station was independent of COVID-19 conditions. However, trip frequency and property ownership were the most important variables in splitting segments before pre-COVID-19 and during COVID-19, respectively. Furthermore, the type of residential and trip purpose was crucial during the COVID-19 period.

**Table 5.** Relevant segmentation of attitudes toward residential location areas and the decision rules for terminal nodes.

| Factor | Node | Level 1 | Level 2 | Level 3 | Level 4 | Level 5 | % N (n) | % Prefer (n) |
|---|---|---|---|---|---|---|---|---|
| 1UrbanArea | 3 | T02; 0, 2, ≥3 cards | R03; Rent, Owner | | | | 65.2 (445) | 51.5 (229) |
| | 4 | T02; 0, 2, ≥3 cards | R03; Hire purchase | | | | 3.8 (26) | 76.9 (20) |
| | 5 | T02; 1 card | T18; 0–50 THB | | | | 7.2 (49) | 51.0 (25) |
| | 7 | T02; 1 card | T18; 51–100, 101–150, >150 THB | S04; Personal Business, Unemployed, Other job | | | 3.1 (21) | 100 (21) |
| | 8 | T02; 1 card | T18; 51–100, 101–150, >150 THB | S04; Student | | | 20.7 (141) | 69.5 (98) |
| 2UrbanArea | 5 | T02; 1 card | S16; Home, Office/Factory | | | | 22.9 (156) | 72.4 (113) |
| | 6 | T02; 1 card | S16; Coffee shop, Other/no, Field site | | | | 8.1 (55) | 49.1 (27) |
| | 9 | T02; 0, 2, ≥3 cards | T03; >1000 m | T02; 0, ≥3 cards | | | 17.0 (116) | 60.8 (74) |
| | 10 | T02; 0, 2, ≥3 cards | T03; >1000 m | T02; 2 cards | | | 2.2 (15) | 26.7 (4) |
| | 7 | T02; 0, 2, ≥3 cards | T03; <400, 400–1000 m | T26; 4–5 times/day | | | 24.9 (170) | 28.8 (49) |
| | 12 | T02; 0, 2, ≥3 cards | T03; <400, 400–1000 m | T26; 2–3, 6–7, 8–9 times/day | T28; 101–150 THB | | 4.0 (27) | 74.1 (20) |
| | 13 | T02; 0, 2, ≥3 cards | T03; <400, 400–1000 m | T26; 2–3, 6–7, 8–9 times/day | T28; 0–50, 51–100, >150 THB | S02; <18, 35–44, 45–54, 55–64 years | 10.2 (70) | 25.7 (18) |
| | 14 | T02; 0, 2, ≥3 cards | T03; <400, 400–1000 m | T26; 2–3, 6–7, 8–9 times/day | T28; 0–50, 51–100, >150 THB | S02; 18–24, 25–34, >64 years | 10.7 (73) | 54.8 (40) |
| 1ResidentialArea | 4 | T18; 0–50 THB | R02; Apartment, Townhouse, Condominium, Other | | | | 17.6 (120) | 61.7 (74) |
| | 5 | T18; 0–50 THB | R02; Single home | | | | 10.7 (73) | 39.7 (29) |
| | 8 | T18; 101–150, >150 THB | R03; Rent, Owner | | | | 19.1 (130) | 70.0 (91) |
| | 9 | T18; 101–150, >150 THB | R03; Hire purchase | | | | 3.1 (21) | 95.2 (20) |
| | 12 | T18; 51–100 THB | T02; 1 card | R01; 2, 4, ≥5 peoples | | | 8.8 (60) | 68.3 (41) |
| | 13 | T18; 51–100 THB | T02; 1 card | R01; 1, 3 peoples | | | 6.5 (44) | 93.2 (41) |
| | 11 | T18; 51–100 THB | T02; 0, 2, ≥3 cards | S04; Employee, Other job | | | 17.3 (118) | 42.4 (50) |
| | 14 | T18; 51–100 THB | T02; 0, 2, ≥3 cards | S04; Personal Business, Unemployed, Student | R01; 1, 3, 4, ≥5 peoples | | 13.5 (92) | 77.2 (71) |
| | 15 | T18; 51–100 THB | T02; 0, 2, ≥3 cards | S04; Personal Business, Unemployed, Student | R01; 2 peoples | | 3.4 (24) | 41.7 (10) |
| 2ResidentialArea | 5 | T02; 1, ≥3 cards | T28; 0–50 THB | | | | 8.1 (55) | 54.5 (30) |
| | 6 | T02; 1, ≥3 cards | T28; 51–100, 101–150, >150 THB | | | | 23.5 (160) | 75 (120) |
| | 4 | T02; 0, 2 cards | T03; >1000 m | | | | 19.1 (130) | 63.1 (82) |
| | 7 | T02; 0, 2 cards | T03; <400, 400–1000 m | R02; Apartment, Single home | | | 37.1 (253) | 43.1 (109) |
| | 9 | T02; 0, 2 cards | T03; <400, 400–1000 m | R02; Townhouse, Condominium, Other | S03; High school, Bachelor, ≥Master | | 8.8 (61) | 52.5 (32) |
| | 10 | T02; 0, 2 cards | T03; <400, 400–1000 m | R02; Townhouse, Condominium, Other | S03; >High school | | 3.4 (23) | 91.3 (21) |

**Table 5.** *Cont.*

| Factor | Node | Level 1 | Level 2 | Level 3 | Level 4 | Level 5 | % N (n) | % Prefer (n) |
|---|---|---|---|---|---|---|---|---|
| 1RuralArea | 3 | T14; 0–3, 4–6, 7–9 times/week | R02; Apartment, Single home, Condominium | | | | 70.7 (482) | 50.0 (241) |
| | 4 | T14; 0–3, 4–6, 7–9 times/week | R02; Townhouse, Other | | | | 18.8 (128) | 67.2 (86) |
| | 5 | T14; ≥10 times/week | S01; Female | | | | 7.6 (52) | 17.3 (9) |
| | 6 | T14; ≥10 times/week | S01; Male | | | | 2.9 (20) | 45.0 (9) |
| 2RuralArea | 2 | R02; Townhouse, Other | | | | | 20.2 (138) | 63.0 (87) |
| | 7 | R02; Apartment, Single home, Condominium | T03; >1000 m | R01; 1, 3, ≥5 peoples | | | 10.3 (70) | 72.9 (51) |
| | 8 | R02; Apartment, Single home, Condominium | T03; >1000 m | R01; 2, 4 peoples | | | 10.1 (69) | 43.5 (30) |
| | 6 | R02; Apartment, Single home, Condominium | T03; <400, 400–1000 m | T24; 4–6, ≥10 times/week | | | 41.2 (281) | 48.4 (136) |
| | 9 | R02; Apartment, Single home, Condominium | T03; <400, 400–1000 m | T24; 0–3, 7–9 times/week | T03; <400 m | | 8.8 (60) | 21.7 (13) |
| | 10 | R02; Apartment, Single home, Condominium | T03; <400, 400–1000 m | T24; 0–3, 7–9 times/week | T03; 400–1000 m | | 9.4 (64) | 43.8 (28) |

**Table 6.** Relevant segmentation of attitudes toward residential accessibility and the decision rules for terminal nodes.

| Factor | Node | Level 1 | Level 2 | Level 3 | Level 4 | Level 5 | Level 6 | % N (n) | % Prefer (n) |
|---|---|---|---|---|---|---|---|---|---|
| 1MassTransit | 2 | T14; 7–9, ≥10 times/week | | | | | | 14.8 (101) | 44.6 (45) |
| | 3 | T14; 0–3, 4–6 times/week | T02; 0, 3 cards | | | | | 53.5 (365) | 67.1 (245) |
| | 5 | T14; 0–3, 4–6 times/week | T02; 1, 2 cards | S01; Female | | | | 18.5 (126) | 92.6 (117) |
| | 6 | T14; 0–3, 4–6 times/week | T02; 1, 2 cards | S01; Male | | | | 13.2 (90) | 77.8 (70) |
| 2MassTransit | 6 | T02; 1, 2 cards | T24; 7–9, ≥10 times/week | | | | | 6.5 (44) | 45.5 (20) |
| | 9 | T02; 1, 2 cards | T24; 0–3, 4–6 times/week | R02; Apartment, Townhouse, Other | | | | 16.7 (114) | 92.1 (105) |
| | 10 | T02; 1, 2 cards | T24; 0–3, 4–6 times/week | R02; Single home, Condominium | | | | 15.5 (106) | 76.4 (81) |
| | 4 | T02; 0, ≥3 cards | T03; >1000 m | | | | | 17.0 (116) | 73.3 (85) |
| | 8 | T02; 0, ≥3 cards | T03; <400, 400–1000 m | R02; Townhouse, Condominium, Other | | | | 10.6 (72) | 68.1 (49) |
| | 12 | T02; 0, ≥3 cards | T03; <400, 400–1000 m | R02; Single home | T01; 1, 2, 3, 4 cars | | | 16.9 (115) | 60.0 (69) |
| | 14 | T02; 0, ≥3 cards | T03; <400, 400–1000 m | R02; Single home | T01; 0 car | T27; 0–30 min | | 1.5 (10) | 90.0 (9) |
| | 15 | T02; 0, ≥3 cards | T03; <400, 400–1000 m | R02; Single home | T01; 0 car | T27; 31–60, 61–90, 91–120, 121–180, ≥180 min | T03; <400 m | 5.0 (34) | 17.6 (6) |
| | 16 | T02; 0, ≥3 cards | T03; <400, 400–1000 m | R02; Single home | T01; 0 car | T27; 31–60, 61–90, 91–120, 121–180, ≥180 min | T03; 400–1000 m | 10.3 (71) | 38.0 (27) |
| 1BusStop | 2 | R02; Single home, Other | | | | | | 44.7 (305) | 60.0 (183) |
| | 4 | R02; Apartment, Townhouse, Condominium | T03; >1000 m | | | | | 15.1 (103) | 86.4 (89) |
| | 5 | R02; Apartment, Townhouse, Condominium | T03; <400, 400–1000 m | T02; 0, ≥3 cards | | | | 22.6 (154) | 64.9 (100) |
| | 7 | R02; Apartment, Townhouse, Condominium | T03; <400, 400–1000 m | T02; 1, 2 cards | T14; 0–3, 4–6 times/week | | | 14.8 (101) | 88.1 (89) |
| | 8 | R02; Apartment, Townhouse, Condominium | T03; <400, 400–1000 m | T02; 1, 2 cards | T14; 7–9, ≥10 times/week | | | 2.8 (19) | 47.4 (9) |
| 2BusStop | 5 | T03; >1000 m | R01; 1, 2, 3, ≥5 peoples | | | | | 22.0 (150) | 81.3 (122) |
| | 6 | T03; >1000 m | R01; 4 peoples | | | | | 4.7 (32) | 56.2 (18) |
| | 9 | T03; <400, 400–1000 m | T02; 1 card | T26; 4–5, 6–7 times/day | | | | 19.2 (131) | 81.7 (107) |
| | 10 | T03; <400, 400–1000 m | T02; 1 card | T26; 2–3, 8–9 times/day | | | | 4.3 (29) | 44.8 (13) |
| | 11 | T03; <400, 400–1000 m | T02; 0, 2, ≥3 cards | S03; High school, College | T28; 0–50 THB | | | 9.8 (67) | 77.6 (52) |
| | 14 | T03; <400, 400–1000 m | T02; 0, 2, ≥3 cards | S03; <High school, Bachelor, ≥Master | S01; Male | | | 7.5 (51) | 58.8 (30) |
| | 16 | T03; <400, 400–1000 m | T02; 0, 2, ≥3 cards | S03; High school, College | T28; 51–100, 101–150, >150 THB | S03; College | | 6.9 (47) | 68.1 (32) |
| | 17 | T03; <400, 400–1000 m | T02; 0, 2, ≥3 cards | S03; <High school, Bachelor, ≥Master | S01; Female | R02; Apartment, Single home, Condominium, Other | | 13.4 (92) | 32.6 (30) |
| | 18 | T03; <400, 400–1000 m | T02; 0, 2, ≥3 cards | S03; <High school, Bachelor, ≥Master | S01; Female | R02; Townhouse | | 1.8 (12) | 83.3 (10) |
| | 19 | T03; <400, 400–1000 m | T02; 0, 2, ≥3 cards | S03; High school, College | T28; 51–100, 101–150, >150 THB | S03; High school | T03; 400 ms | 3.7 (25) | 64.0 (16) |
| | 20 | T03; <400, 400–1000 m | T02; 0, 2, ≥3 cards | S03; High school, College | T28; 51–100, 101–150, >150 THB | S03; High school | T03; 400–1000 m | 6.7 (46) | 34.8 (16) |

**Table 6.** *Cont.*

| Factor | Node | Level 1 | Level 2 | Level 3 | Level 4 | Level 5 | Level 6 | % N (n) | % Prefer (n) |
|---|---|---|---|---|---|---|---|---|---|
| 1Highway | 1 | T14; 0–3, 7–9, ≥10 times/week | | | | | | 35.2 (240) | 38.8 (93) |
| | 3 | T14; 4–6 times/week | T02; 0, 2 cards | | | | | 41.60 (284) | 52.8 (150) |
| | 5 | T14; 4–6 times/week | T02; 1, ≥3 cards | T18; 0–50, >150 THB | | | | 6.7 (46) | 87.0 (40) |
| | 6 | T14; 4–6 times/week | T02; 1, ≥3 cards | T18; 51–100, 101–150 THB | | | | 16.5 (112) | 61.6 (69) |
| 2Highway | 3 | R03; Rent, Owner | T24; 0–3, 7–9, ≥10 times/week | | | | | 37.5 (256) | 42.4 (108) |
| | 5 | R03; Hire purchase | T28; 0–50, 101–150, >150 THB | | | | | 4.4 (30) | 90.0 (27) |
| | 6 | R03; Hire purchase | T28; 51–100 THB | | | | | 2.9 (20) | 55.0 (11) |
| | 9 | R03; Rent, Owner | T24; 4–6 times/week | T02; 0, 2, ≥3 cards | TP2; SW + HM, PB + HM, VS + HM, SW + SH + HM, SH + SW + HM | | | 34.3 (234) | 45.7 (107) |
| | 10 | R03; Rent, Owner | T24; 4–6 times/week | T02; 0, 2, ≥3 cards | TP2; SH + HM, OT + HM, SW + PB + HM, PB + SW + HM, PB + HM + SH + HM | | | 2.9 (20) | 95.0 (19) |
| | 11 | R03; Rent, Owner | T24; 4–6 times/week | T02; 1 card | T28; 0–50, >150 THB | | | 5.5 (37) | 83.8 (31) |
| | 12 | R03; Rent, Owner | T24; 4–6 times/week | T02; 1 card | T28; 51–100, 101–150 THB | | | 12.5 (85) | 56.5 (48) |

*4.3. Segmentation by Attitude toward Concern of COVID-19*

The attitude tree of attitudes toward concern about COVID-19 was constructed for attitudes toward not choosing to live in an urban area, or regarding the use of public transport due to concern about infection. The results of each attitude decision tree are described as follows:

The CHAID analysis results of attitudes toward not choosing to live in an urban area due to concern about infection by pre-COVID-19 consisted of nine nodes, three levels, five terminal nodes, and two branches classified by the number of transport cards owned (T02), which represented the most important variables. Terminal 3 had the highest proportion of respondents agreeing to prefer not to live in an urban area due to concern about infection (49.4 percent of respondents) and agreeing with 49.3% attitude. The segment decision rule is sorted at level 1 by variable T02 (0 and 2 cards) and at level 2 by variable T03 (<400 and 400–1000 m). During the COVID-19 period, the tree result consisted of seven nodes, three levels, four terminal nodes, and two branches classified by walking distance to the nearest station (T03), which is the most important variable in the decision tree. The highest proportion was demonstrated by 73.3%of respondents in terminal 1 of level 1 by variable T03 (<400 and 400–1000 m), who agreed with 57.2% of the attitude. Nevertheless, the difference in the decision tree showed that the type of residential pre-COVID-19 was an important variable, whereas travel time was an important variable during COVID-19, as shown in Table 7 and Figure 4. Based on the validation sample of the decision tree technique, the CHAID algorithm had an accuracy of 59.1% before and 62.0% during COVID-19.

**Table 7.** Relevant segmentation of attitude toward not choosing to live in an urban area due to concern about infection and decision rule for terminal node.

| Factor | Node Number | Level 1 | Level 2 | Level 3 | % N (n) | % Agree (n) |
|---|---|---|---|---|---|---|
| | 3 | T02; 0, 2 cards | T03; <400, 400–1000 m | | 49.4 (337) | 49.3 (166) |
| | 5 | T02; 1, ≥3 cards | R02; Apartment | | 11.0 (75) | 52.0 (39) |
| 1UrbanConcern | 6 | T02; 1, ≥3 cards | R02; Single home, Townhouse, Condominium, Other | | 20.5 (140) | 72.1 (101) |
| | 7 | T02; 0, 2 cards | T03; >1000 m | T02; 0 card | 16.9 (115) | 69.6 (80) |
| | 8 | T02; 0, 2 cards | T03; >1000 m | T02; 2 cards | 2.2 (15) | 80.0 (12) |
| | 1 | T03; <400, 400–1000 m | | | 73.3 (500) | 57.2 (286) |
| | 4 | T03; >1000 m | T02; 2 cards | | 2.2 (15) | 20.0 (3) |
| 2UrbanConcern | 5 | T03; >1000 m | T02; 0, 1, ≥3 cards | T27; 0–30, 91–120, 121–180, ≥180 min | 12.2 (83) | 62.7 (52) |
| | 6 | T03; >1000 m | T02; 0, 1, ≥3 cards | T27; 31–60, 61–90 min | 12.3 (84) | 86.9 (73) |

The CHAID analysis results of attitude toward infection concerns to use public transport pre-COVID-19 consisted of 11 nodes, four levels, six terminal nodes, and two branches classified by trip frequency (T14), which were the most important variables. The highest proportion is illustrated for terminal 8 on the segment decision rule of level 1 by variable T14 (0–3 and 4–6 times/week), level 2 by variable R03 (rent and owner), level 3 by variable S16 (home, office/factory, coffee shop, and co-working space), and level 4 by variable R03 (owner), as represented by 26.0% of respondents and agreeing with 62.7% of the attitude toward concern about contracting an infection from the use of public transport. The tree result during the COVID-19 period consisted of nine nodes, four levels, five terminal nodes, and two branches classified by type of residence (R02), which was the most important variable. The segment of terminal 3 had the highest proportion (59.4%) of respondents and 54.6% agreed with the attitude and segment decision rule shown on level 1 by variable R02 (apartment, single home, and condominium) and level 2 by variable T03 (<400 and 400–1000 m). Nevertheless, the difference in the decision tree showed that property owner-

ship, place of work, and gender in the pre-COVID-19 period were the important variables, whereas the type of residence, walking distance to the nearest station, and the number of transport card ownership became important variables during the COVID-19 period, as shown in Table 8 and Figure 5. The CHAID algorithm of attitude toward concern about contracting from using public transport had an accuracy of 62.3 and 63.9% before and after COVID-19, respectively.

The CHAID decision tree was used to determine the segmentation characteristic profile of travelers and residences in the vicinity of mass transit stations with the highest accessibility of travel modes. The CHAID model provided segmentation of the relationship between independent and attitude dependent variables. Gender, place of work, number of transport card ownerships, walking distance to the nearest station, type of residence, property ownership, trip frequency, and travel cost are among the variables having the same correlation in all models of the pre-COVID-19 and COVID-19 periods.

Furthermore, prior to COVID-19, occupation variables were found to have influenced attitudes toward preferring to live in urban areas and residential areas. However, during the COVID-19 period, the following variables were related to attitude: age on preferring to live in urban areas, education on preferring to live in residential areas and preferring residential areas near bus stops, number of vehicles owned in preferring residential areas near mass transit stations, number of transfers in preferring to live in urban areas and preferring residential areas near bus stops, travel time in preferring residential areas near mass transit stations and not choosing to live in an urban area due to concern about infection attitude, and trip purpose on preferring residential areas near highways or main roads. Table 9 provides a summary of the model, describing the important variables by segment for all decision trees pre and during COVID-19, as well as the accuracy demonstrated by the model.

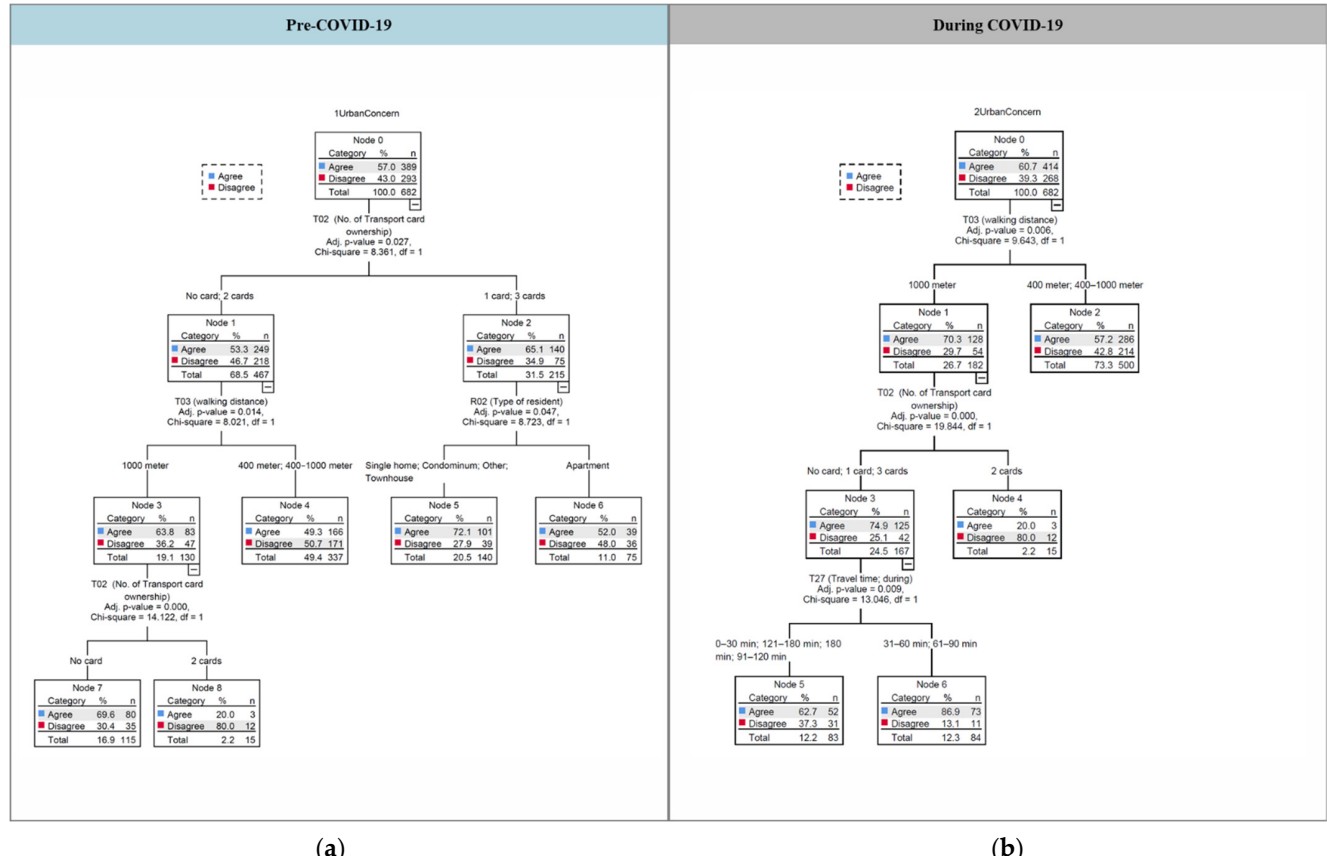

**Figure 4.** Decision tree map of attitude toward not choosing to live in an urban area due to concern about infection for (**a**) pre-COVID-19 and (**b**) during COVID-19.

**Table 8.** Relevant segmentation of attitude toward concern about contracting infection from use of public transport and decision rule for terminal node.

| Factor | Node Number | Level 1 | Level 2 | Level 3 | Level 4 | % N (n) | % Agree (n) |
|---|---|---|---|---|---|---|---|
| 1PTconcern | 2 | T14; 7–9, ≥10 times/week | | | | 14.8 (101) | 24.8 (25) |
| | 4 | T14; 0–3, 4–6 times/week | R03; Hire purchase | | | 6.9 (47) | 87.2 (41) |
| | 7 | T14; 0–3, 4–6 times/week | R03; Rent, Owner | S16; Home, Office/Factory, Coffee shop, Co-working space | R03; Rent | 26.1 (178) | 51.1 (91) |
| | 8 | T14; 0–3, 4–6 times/week | R03; Rent, Owner | S16; Home, Office/Factory, Coffee shop, Co-working space | R03; Owner | 26.0 (177) | 62.7 (111) |
| | 9 | T14; 0–3, 4–6 times/week | R03; Rent, Owner | S16; Other/no, Field site | S01; Female | 17.2 (117) | 46.2 (54) |
| | 10 | T14; 0–3, 4–6 times/week | R03; Rent, Owner | S16; Other/no, Field site | S01; Male | 9.1 (62) | 30.6 (19) |
| 2PTconcern | 2 | R02; Townhouse, Other | | | | 20.2 (138) | 79.0 (109) |
| | 3 | R02; Apartment, Single home, Condominium | T03; <400, 400–1000 m | | | 59.4 (405) | 54.6 (221) |
| | 6 | R02; Apartment, Single home, Condominium | T03; >1000 m | T24; ≥10 times/week | | 3.1 (21) | 33.3 (7) |
| | 7 | R02; Apartment, Single home, Condominium | T03; >1000 m | T24; 0–3, 4–6, 7–9 times/week | T02; 0, 1, ≥3 cards | 15.7 (107) | 79.4 (85) |
| | 8 | R02; Apartment, Single home, Condominium | T03; >1000 m | T24; 0–3, 4–6, 7–9 times/week | T02; 2 cards | 1.6 (11) | 36.4 (4) |

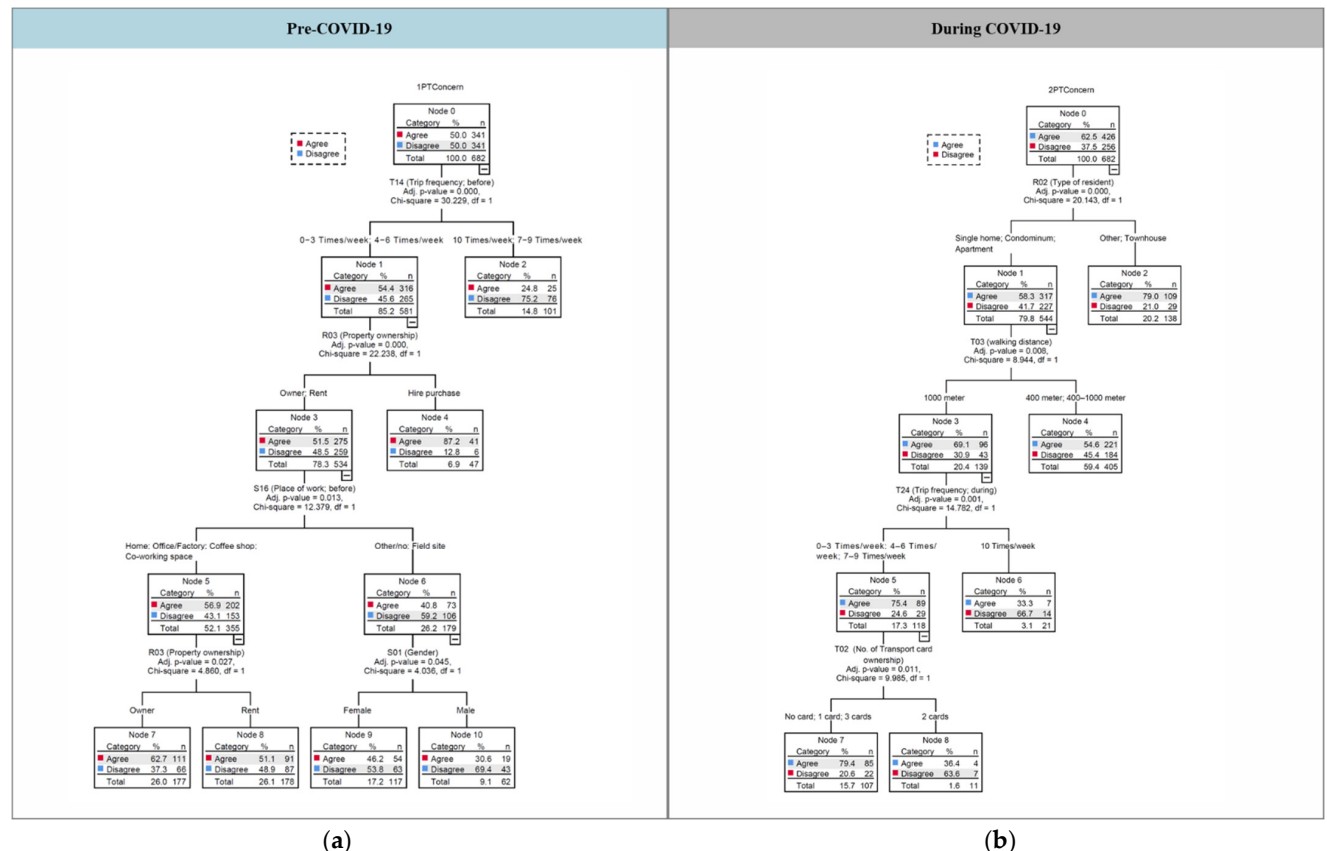

(**a**) (**b**)

**Figure 5.** Decision tree map of attitude toward concern about contracting infection from use of public transport for (**a**) Pre-COVID-19 (**b**) During COVID-19.

**Table 9.** Summary of the node level and *p*-value of relevant variables on attitudes.

| Independent Variables Pre-COVID-19 | Attitude toward Residential Location Area | | | Attitude toward Residential Accessibility | | | Attitude toward Concern of COVID-19 | |
|---|---|---|---|---|---|---|---|---|
| | 1Urban Area | 1Residential Area | 1Rural Area | 1Mass Transit | 1Bus Stop | 1Highway | 1Urban Concern | 1PT Concern |
| S01 | | | 2 (0.015) | 3 (0.001) | | | | 4 (0.045) |
| S04 | 3 (0.047) | 3 (0.000) | | | | | | |
| S16 | | | | | | | | 3 (0.013) |
| T02 | 1 (0.001) | 2 (0.000) | | 2 (0.000) | 3 (0.015) | 2 (0.007) | 1 (0.027), 3 (0.000) | |
| T03 | | | | | 2 (0.012) | | 2 (0.014) | |
| R01 | | 3 (0.033), 4 (0.011) | | | | | | |
| R02 | | 2 (0.046) | 2 (0.008) | | 1 (0.000) | | 2 (0.047) | |
| R03 | 2 (0.034) | 2 (0.045) | | | | | | 2 (0.000), 4 (0.027) |
| T14 | | | 1 (0.000) | 1 (0.000) | 4 (0.000) | 1 (0.000) | | 1 (0.000) |
| T18 | 2 (0.022) | 1 (0.004) | | | | 3 (0.012) | | |
| Overall correct | 57.6% | 68.0% | 55.9% | 71.6% | 69.1% | 59.5% | 59.1% | 62.3% |
| Number of nodes | 9 | 16 | 7 | 7 | 9 | 7 | 9 | 11 |
| Number of terminals | 5 | 9 | 4 | 4 | 5 | 4 | 5 | 6 |
| Independent Variables During COVID-19 | Attitude toward Residential Location Area | | | Attitude toward Residential Accessibility | | | Attitude toward Concern of COVID-19 | |
| | 2Urban Area | 2Residential Area | 2Rural Area | 2Mass Transit | 2Bus Stop | 2Highway | 2Urban Concern | 2PT Concern |
| S01 | | | | | 4 (0.017) | | | |
| S02 | 5 (0.025) | | | | | | | |
| S03 | | 4 (0.015) | | | 3 (0.018), 5 (0.018) | | | |
| S26 | 2 (0.024) | | | | | | | |
| T01 | | | | 4 (0.006) | | | | |
| T02 | 1 (0.000), 3 (0.018) | 1 (0.000) | | 1 (0.000) | 2 (0.000) | 3 (0.040) | 2 (0.000) | 4 (0.011) |
| T03 | 2 (0.000) | 2 (0.011) | 2 (0.009), 4 (0.009) | 2 (0.000), 6 (0.035) | 1 (0.000), 6 (0.018) | | 1 (0.006) | 2 (0.008) |
| R01 | | | 3 (0.007) | | 2 (0.033) | | | |
| R02 | | 3 (0.022) | 1 (0.016) | 3 (0.050), 3 (0.019) | 5 (0.010) | | | 1 (0.000) |
| R03 | | | | | | 1 (0.001) | | |
| T24 | | | 3 (0.029) | 2 (0.000) | | 2 (0.024) | | 3 (0.001) |
| T26 | 3 (0.008) | | | | 3 (0.000) | | | |
| T27 | | | | 5 (0.007) | | | 3 (0.009) | |
| T28 | 4 (0.009) | 2 (0.031) | | | 4 (0.011) | 2 (0.032), 4 (0.026) | | |
| TP2 | | | | | | 4 (0.012) | | |
| Overall correct | 67.3% | 62.9% | 59.4% | 72.4% | 72.6% | 60.3% | 62.0% | 63.9% |
| Number of nodes | 15 | 11 | 11 | 17 | 21 | 13 | 7 | 9 |
| Number of terminals | 8 | 6 | 6 | 9 | 11 | 7 | 4 | 5 |

(*p*-value).

## 5. Hypothesis Testing

SEM approach was used to investigate the determinants of change in attitudes due to COVID-19 concerns toward residential location areas, and toward residential accessibility. The pre-test and post-test designs, and the first-order factor model were applied to the test model to hypothesize the relationship influenced by COVID-19.

The intervention factors were defined by the COVID-19 concern attitude change, with the model divided into two models along the dimensions of (1) attitude toward residential location area and (2) attitude toward residential accessibility. For each model, four latent variables representing pre-COVID-19 and COVID-19 periods were defined. The hypothesis of the study is that attitude toward residential accessibility related to travel mode will influence attitudes toward residential location areas because the type of residential location has an effect on travel behavior [19].

## *5.1. Goodness-of-Fit*

An analysis of SEM was conducted using the AMOS 23.0 software package. A maximum likelihood estimator was utilized, and 5000 bootstrap samples were used to obtain the bias-corrected confidence intervals for each parameter. Bootstrapping is a resampling method in which the original sample is considered representative of the population [52].

The indicated model results, based on the recommended goodness of fit, are shown in Table 10. The chi-square value was significant at the 0.000 significance level. The chi-square divided by the number of degrees of freedom was higher than the acceptance value of four, suggesting an acceptable fit. The root mean square error of approximation (RMSEA) value was greater than the expected value of acceptable fit (0.07), and the comparative fit index (CFI), the goodness of fit index (GFI), and Tucker–Lewis index (TLI) values were greater than the acceptable goodness-of-fit cutoff score of 0.90. The goodness-of-fit test showed that all models fit adequately and are statistically significant.

**Table 10.** Recommended fitness index and results of the model.

| Index | Level of Acceptance | Model Result |
|---|---|---|
| Chi-square/df [53] | 1–4 | 3.289 |
| *p*-value | <0.05 | 0.000 |
| RMSEA [54] | <0.07 | 0.058 |
| GFI [55] | ≥0.90 | 0.960 |
| CFI [56] | ≥0.90 | 0.961 |
| TLI [56] | ≥0.90 | 0.943 |

## *5.2. SEM Model Results*

The model was evaluated for attitudes toward residential location areas and accessibility, as well as whether their relationship was affected by the COVID-19 pandemic. The results of structural equation modeling revealed a significant influence of attitudes toward residential accessibility, with a relationship of 0.794 between pre-COVID-19.

(Pre-COVID Accessibility) and during COVID-19 (During COVID Accessibility). The intervention variables of 2PTconcern were affected by During COVID Accessibility with a value of 0.075. The variance of the dependent variables or squared multiple correlations ($R^2$) of During COVID Accessibility affected by Pre-COVID Accessibility and the intervention variable of 2PTconcern explained is 65% of During COVID Accessibility as shown in Figure 6. The relationship of attitudes toward residential location areas shows that pre-COVID-19 (Pre-COVID Location) to during COVID-19 (During COVID Location) had a positive value of 0.464. The intervention variables of 2UrbanConcern were affected by During COVID Location with a value of 0.075. Pre-COVID Location and 2UrbanConcern explained 82 percent of the effect of During COVID Location. The results of the research hypothesis study found that the relationship between During COVID-19 Accessibility and During COVID-19 Location indicated a significant positive relationship and had a direct significant influence on During COVID Location (0.514). Table 11 and Figure 6 show the standardized path coefficients of the structural model.

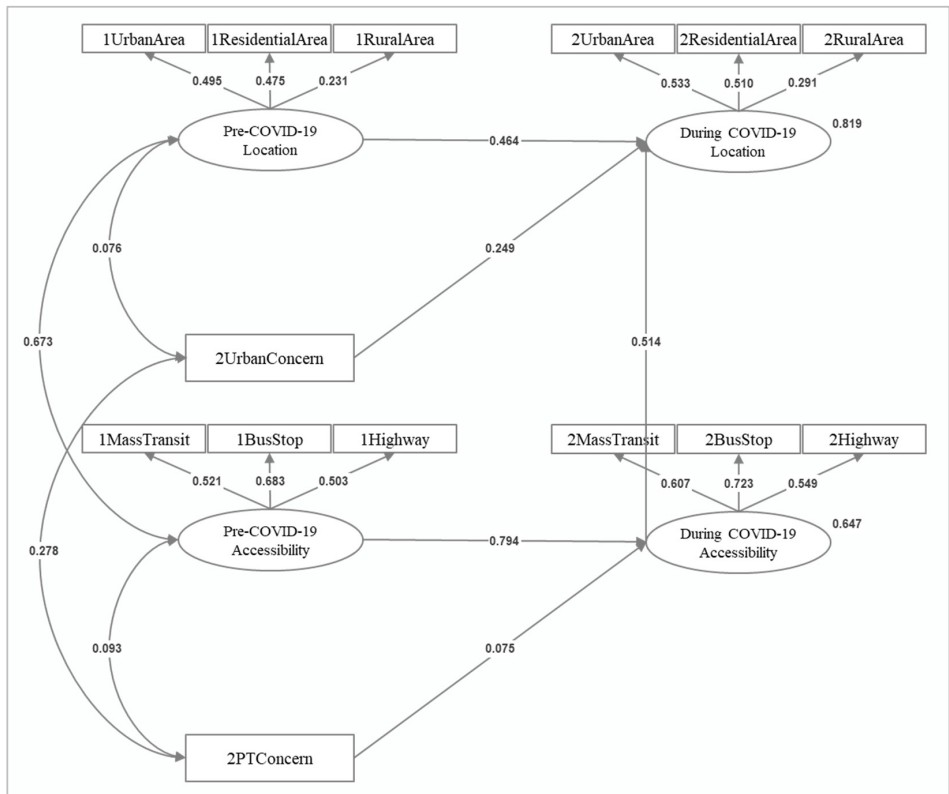

**Figure 6.** Pre-test and post-test model specification and standardized estimates.

**Table 11.** Parameter estimates of regression weight and correlation of model result.

| Regression Paths | β | p |
|---|---|---|
| Pre-COVID Location | | |
| 1UrbanArea | 0.495 | *** |
| 1ResidentialArea | 0.475 | *** |
| 1RuralArea | 0.231 | *** |
| During COVID Location | 0.464 | |
| During COVID Location | | |
| 2UrbanArea | 0.533 | *** |
| 2ResidentialArea | 0.510 | *** |
| 2RuralArea | 0.291 | *** |
| 2UrbanConcern | | |
| During COVID Location | 0.249 | *** |
| Pre-COVID Accessibility | | |
| 1MassTransit | 0.521 | *** |
| 1BusStop | 0.683 | *** |
| 1Highway | 0.503 | *** |
| During COVID Accessibility | 0.794 | *** |
| During COVID Accessibility | | |
| 2MassTransit | 0.607 | *** |
| 2BusStop | 0.723 | *** |
| 2Highway | 0.549 | *** |
| During COVID Location | 0.514 | *** |
| 2PTConcern | | |
| During COVID Accessibility | 0.075 | 0.021 |
| **Correlation paths** | **β** | **p** |
| Pre-COVID Location | | |
| 2UrbanConcern | 0.076 | 0.045 |
| Pre-COVID Accessibility | 0.673 | *** |
| Pre-COVID Accessibility | | |
| 2PTconcern | 0.093 | 0.035 |
| 2PTConcern | | |
| 2UrbanConcern | 0.278 | *** |

*** Significant at the 0.001.

## 6. Discussion

Residential self-selection can lead to relocation related to travel behavior and various variables (such as sociodemographic characteristics, residential characteristics, and travel characteristics). Furthermore, to understand future effects, travel-related attitudes were significant predictors of their travel evaluation [17] and motivations for relocation and discovered that the reasons for relocation were travel-related [6]. However, the uncertain situation of COVID-19 directly affected the behavior and attitude toward relocation in this study. The characteristics of the study area showed that mass transit and feeder transit were the main modes of transport used by people. Traveling by non-motorized and paratransit vehicles increased slightly during the COVID-19 pandemic, similar to previous research [1]. During COVID-19, passengers were more concerned about public transportation usage than they were before COVID-19 [38], which may impact housing type preference [41], as shown by the change in attitude toward residential location areas.

The study of the segmentation of travelers and residents around mass transit station areas has qualifying variables to understand the characteristics of the travelers and residents under consideration for attitude-based relocation related to travel behavior. The decision tree identified variables of age, education, number of car ownership, number of transfers, travel time, and travel cost of significant importance to consider compared with than the pre-COVID-19 period. Evidently, during the COVID-19 pandemic, people concentrated on travel time, decreasing the number of transfers, and eliminating unnecessary travel purposes. Consistent with past pandemics, MERS reduced trips during the pandemic [57]. Additionally, the segmentation results further confirmed that the most significant variables relating to traveler and resident characteristics were the number of transport cards owned, walking distance to the nearest mass station, number of households, type of resident, property ownership, travel cost, and trip frequency.

As a result, the hypotheses of the study that are based on attitudes toward residential accessibility in relation to travel modes will influence attitudes toward residential location areas. The SEM results revealed that attitudes toward residential accessibility of travel mode were a significant determinant of attitudes toward residential location areas, thereby supporting residential self-selection or relocation based on the attitude hypothesis for normal situations (pre-COVID) and pandemic situations (during COVID-19). However, it is not surprising that the pre-COVID-19 latent variable had a direct effect on the variables during COVID-19. The intervention variable of concern for using public transport had a slight effect during COVID-19 on the accessibility of travel modes, whereas the variable of concern for living in an urban area had a stronger effect during COVID-19 on the location area.

## 7. Conclusions

This study examined the impact of COVID-19 on the majority of traveler and resident characteristic groups in the vicinity of a mass transit station, with the objective of understanding the target of the user and providing information to encourage increased use of mass and feeder transit services, as well as non-motorized transportation under the pandemic situation in the future. However, the allocation of areas, access to mass transit and feeder transit, and neighborhoods that will support the growth of the city, as well as urban development, should improve more appropriately in the future under the trend of residents considering relocation that has been influenced by changes in attitudes and behaviors.

The CHAID has been designed to accommodate a variety of data types, including scale data (also known as continuous data) and categorical data (ordinal or nominal variables). This methodology is well-suited for examining large, complex datasets because it is effective at identifying relationships between independent and dependent variables. The attitudes of the various segments of travelers and resident characteristic groups could help to understand and address any potential differences in pandemic-related travel impacts. The CHAID results could explain the fundamentals of travelers' and residents' characteristics.

Regarding hypothesis analysis, SEM was used to determine the relationship between variables and has confirmed a significant relationship between pre-COVID-19 and during COVID-19 periods and the impact of COVID-19 effects.

Considering the significance of this study, policymakers should place additional emphasis on relocation because of changes in attitudes. It has been demonstrated that the people who live near mass transit stations, within 400 m and between 400–1000 m from the stations, prefer to live in residential and rural areas in the future by 23% and 25%, respectively, while the proportion of people who prefer to live in urban areas is 18%. This reflects people who prefer to avoid commuting by public transportation (feeder transit and mass transit), as evidenced by their attitudes toward residential accessibility of travel modes.

However, according to the CHAID analysis, a limited sample size for analysis was a limitation of the study; a large sample size produced a stronger classification [58]. In the future, synthetic data should be thought of in addition to model validation and evaluating the prediction performance of tree classifiers. Therefore, the preferences and attitudes of decision-makers regarding relocation were considered in this study. However, in the COVID pandemic scenario, a longer forecasting period is required. Additionally, tracking changes in population relocation and using longitudinal data are advantageous for more accurate forecasting.

**Author Contributions:** Conceptualization, C.P. and K.S.; methodology, K.H.; software, C.P.; validation, C.P., and K.S.; formal analysis, C.P.; investigation, C.P.; resources, K.S.; data curation, C.P.; writing—original draft preparation, C.P.; writing—review and editing, C.P.; visualization, K.H.; supervision, K.S. All authors have read and agreed to the published version of the manuscript.

**Funding:** This research received no external funding.

**Institutional Review Board Statement:** Not applicable.

**Informed Consent Statement:** Not applicable.

**Data Availability Statement:** Not applicable.

**Conflicts of Interest:** The authors declare no conflict of interest.

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
