# Peer review of "Attitude-Based Segmentation of Residential Self-Selection and Travel Behavior Changes Affected by COVID-19"

_futuretransp, doi:10.3390/futuretransp2020030_

Round 1

Reviewer 1 Report

Dear author,

Thank you for the opportunity to evaluate your manuscript called “Attitude-based Segmentation of Residential Self-selection and 2 Travel Behavior Changes Affected by COVID-19“.I must say, that your text is generally well written thus I have no major comments on the general structure of the article. Only a few details, mostly of formal or explanatory nature, should be fixed to improve your manuscript:

General – please add analysis of representativeness of the interviewed sample. Obviously, the sample will not be fully representative, but it would be useful to add a column to Table 1 with official percentages of demographic and residential characteristics - ideally for the entire Bangkok metropolitan area

Line 280 onwards – please add aprox. conversion to USD when referring to price/income in THB.

Line 283-285 – incomes mentioned are per annum? Please specify in text. Also short note on average income in Thailand will be helpful as many international readers are not familiar with the situation in country thus the amounts quoted they cannot automatically put into a broader context

Line 293 onwards, table 3 – the term “trip” is referring to one-way or return journey? I recommend to have better description for unit in “Category” column for T14 – from times / week to trips / week

Table 3 – variable “Travel time” refer to the amount of time spent during all trips in day?

Line 316 – how do you know that trip reduction was because of preventing of infection? One may assume this as the main reason, however, direct motivation for travel behaviour change was not part of the questionnaire. Therefore, this statement should belong to the discussion rather than among the results. dtto line 327

Line 348 – please check numbers in text. I cannot believe that using of non-motorised modes changed from 1% to 47 % because of pandemic. These numbers are not in coherence with Fig. 3 where increase from 2 % to 3 % is presented.

Line 676-792 – fix format of references to official MDPI’s style

My final decision is for minor changes

with kind regards

RW

Author Response

Dear Reviewer,

We sincerely appreciate your thoughtful feedback, which assisted us in making improvements to the manuscript. We are certain that all your suggestions have been considered carefully in the revised manuscript.

Reviewer 2 Report

The paper studies the effects of the pandemic on residential mobility patters. In this paper, chi-square automatic interaction detection algorithm was applied owing to its flexibility around categorical variables and its ability for multi-way nodal split. A causal relationship is then evaluated using Structural Equation Modeling between land use and travel patterns on classified based on weekdays work travel and weekend travels.

I would suggest the authors to use other causal modeling techniques in addition to SEM such as Peter Clarks or Rubin causal methodology which would help identify and outline linear relationships better.

Also, it would be interesting to see an intervention model based on a "what if" scenario where instead of decision trees, various influential variables are used. In addition to that, I would suggest using synthetic dataset for validation as well as to strengthen the hypothesis.

Author Response

(The authors gave the same response as above.)

Round 2

Reviewer 2 Report

Thank you to the authors for the response to comments.

I believe the paper is good for publishing.